# Evaluation of Suitable Mixture of Water and Air for Processing Tomato in Drip Irrigation in Xinjiang Oasis

**Chilin Wei [1,2], Yan Zhu [1,2], Jinzhu Zhang [1,2] and Zhenhua Wang [1,2,*]**

1   College of Water Resources and Architectural Engineering, Shihezi University, Shihezi 832000, China;
    wcl285319@163.com (C.W.); zhuyan2020@shzu.edu.cn (Y.Z.); xjshzzjz@shzu.edu.cn (J.Z.)
2   Key Laboratory of Modern Water-Saving Irrigation of Xinjiang Production and Construction Corps,
    Shihezi University, Shihezi 832000, China
*   Correspondence: wzh2002027@163.com

**Abstract:** Aerated irrigation (AI) has emerged as a method to mitigate rhizosphere hypoxia caused by wetting front with sub-surface drip irrigation (SDI). Increasing oxygen in processing tomato's root zone is beneficial to the improvement of the rhizosphere gas environment, crop growth, yield and quality. The relationship between aerated irrigation and irrigation quantity is not clear. A total of eight treatments, including four irrigation levels (4950 m$^3$ hm$^{-2}$ (W1), 4750 m$^3$ hm$^{-2}$ (W2), 4500 m$^3$ hm$^{-2}$ (W3), 4050 m$^3$ hm$^{-2}$ (W4)) in combination with aerated irrigation (A2) and non-aerated irrigation (A1) were used to investigate the effects of aerated irrigation on the physiological characteristics and yield of processing tomatoes under mulched drip irrigation in Xinjiang, China. The effects of aerated irrigation on plant height, stem diameter, leaf area index and dry matter, photosynthesis, fluorescence, fruit quality and yield of processing tomatoes were studied. The results showed that plant height, stem diameter, biomass accumulation and leaf area index of processing tomatoes under aerated irrigation were increased by 10.2%, 7.3%, 12.5% and 6.2% under the W1, W2, W3 and W4 conditions ($p < 0.05$), respectively, compared with non-aerated irrigation. Yield and the content of Vitamin C and soluble solids under aerated irrigation was 9.71%, 5.59% and 5.68% ($p < 0.05$) higher than that under conventional irrigation, respectively, and the sugar-acid under aerated irrigation decreased by 0.5%. Through principal component analysis, W2A2 treatment had a higher score according to the yield index (per fruit weight, fruit number per plant) and quality index (Vitamin C, soluble solids, sugar-acid ratio) than the other treatments. The results show that aerated irrigation is feasible under the existing mulched drip irrigation in Xinjiang and, in this experiment, W2A2 treatment was the most suitable planting mode.

**Keywords:** mulched drip irrigation; aerated irrigation; processing tomato; yield; fruit quality

## 1. Introduction

Drought is one of the major threats to agricultural production and development [1]. Xinjiang, China is a typical arid area with a large temperature difference between day and night. The annual precipitation of Xinjiang is 147 mm, and the annual average evaporation is as high as 1500–2300 mm [2,3]. Water resources in this area are exceedingly scarce, and the sustainable development of agriculture in this area depends on irrigation [4,5]. Therefore, appropriate irrigation strategies should be adopted to save irrigation water, maintain relative yield and improve water use efficiency [6,7], and it is urgent to realize the effective use of water resources [8]. Since the introduction of mulched drip irrigation by Xinjiang Production and Construction Corps in 1996, mulched drip irrigation has been promoted and applied in Xinjiang for 25 years [9]. The mulched drip irrigation technology is a combination of drip irrigation technology and mulched planting technology to distribute irrigation water evenly to the soil and provide appropriate water and nutrients for crops in time [10]. As the most effective irrigation method in Xinjiang, China, the mulched drip irrigation can keep the soil water use efficiency at about 95% in the root zone of crops

without destroying the soil structure. Under the condition of mulched drip irrigation, and comprehensive maintenance of the soil's internal environment, the crops can maintain a good growth state, which is conducive to the absorption of water by crop roots [11,12]. However, the irrigation frequency of mulched drip irrigation is relatively high, and the use of a dripper often leads to the formation of continuous saturated wet zones in the nearby soil due to irrigation [13,14]. As a result, the roots of mulched drip irrigation crops generally tend to grow in saturated wet zones [15,16], therefore the roots of mulched drip irrigation crops often induce soil hypoxia [17]. The studies of Bhattarai et al. [18] and Mchugh et al. [19] have shown that crop yield does not increase with the increase of irrigation water when irrigation water is larger than crop water demand to a certain extent. Payero et al. [20] believe that the mismatch between irrigation and crop yield is due to insufficient oxygen in the soil. The continuous saturated humid area in the soil under the mulched drip irrigation causes the air in the soil to be replaced by water, thereby limiting the availability and mobility of oxygen in the soil pore [21,22].

Aerated irrigation is based on mulched drip irrigation; venturi equipment is used to suction air into drip irrigation pipe, and then the mixture of air and water containing oxygen in the gaseous and dissolved phases [23] is transported to the soil in the root area of crops. The growth and development of crops are affected by many factors, such as water, fertilizer, gas, heat and so on. For a long time, the research has been mainly focused on the effects of water and fertilizer coupling regulation on crop growth and yield. Now, the research on aerated irrigation is becoming more and more common, focusing on water, fertilizer and air, together. Bhattarai et al. [24] found that aerated irrigation could significantly improve fruit yield and water use efficiency in heavy clay and salt-alkali soil, and that aerated irrigation had different effects on the yield of different crops. Li et al. [25] found that root zone ventilation could significantly promote the growth of plant height and stem diameter. Bhattarai et al. [26] found that aerated subsurface irrigation water gives growth and yield benefits to Zucchini, vegetable soybean and cotton. Chen et al. [27] found that the coupling of water, fertilizer and air could promote the growth of Cucumis sativus in greenhouse. Therefore, aerated irrigation can not only achieve higher irrigation water use efficiency under mulched drip irrigation, but can also effectively alleviate hypoxia symptoms in crops under mulched drip irrigation.

In this study, we selected processing tomatoes as research objects and venturi air injector gas processing was used in the drip irrigation head. Through observation and analysis of the growth, yield and quality indexes of the processing tomatoes, the aerated irrigation effects on processing tomato were attained. A certain theoretical basis and technical reference for aerobic irrigation mode and high-yield cultivation was provided.

## 2. Materials and Methods

### 2.1. Study Area

The experiment was conducted from 6 May to 22 August, 2019 in the Key Laboratory of Modern Water-saving Irrigation, Shihezi University, Shihezi city, Xinjiang (85.9° E, 44.3° N, elevation is 412 m). This region is located in the central plain of the Manas River irrigation area, adjacent to the Gurbantunggut Desert, which has a typical temperate continental desert climate (Figure 1). The average temperature during the whole monitoring period in 2019 is 18.6 °C, and the average precipitation is 170 mm, of which 60% is concentrated from May to August (Figure 2). The average annual potential evaporation capacity is 1890 mm, and the summer relative humidity is between 30~50%. The annual sunshine duration is 2447.9 h. The frost-free period is 182 d. The volume mass and field water holding rate (mass moisture content) of 0–100 cm soil layer were 1.60 g/cm$^3$ and 18.65% respectively. The basic physical and chemical properties of soil in the experimental area are shown in Table 1, and the salinity of irrigation water is about 1.35 g/L.

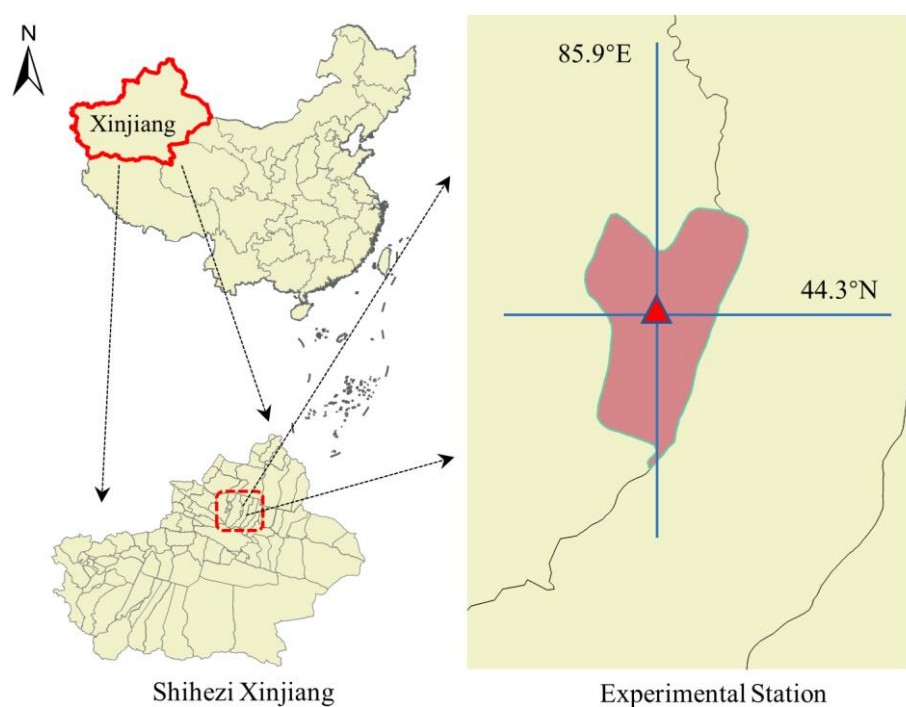

**Figure 1.** Maps and imagery of the study site, the Key Laboratory of Modern Water-saving Irrigation, Shihezi University, Shihezi city, Xinjiang.

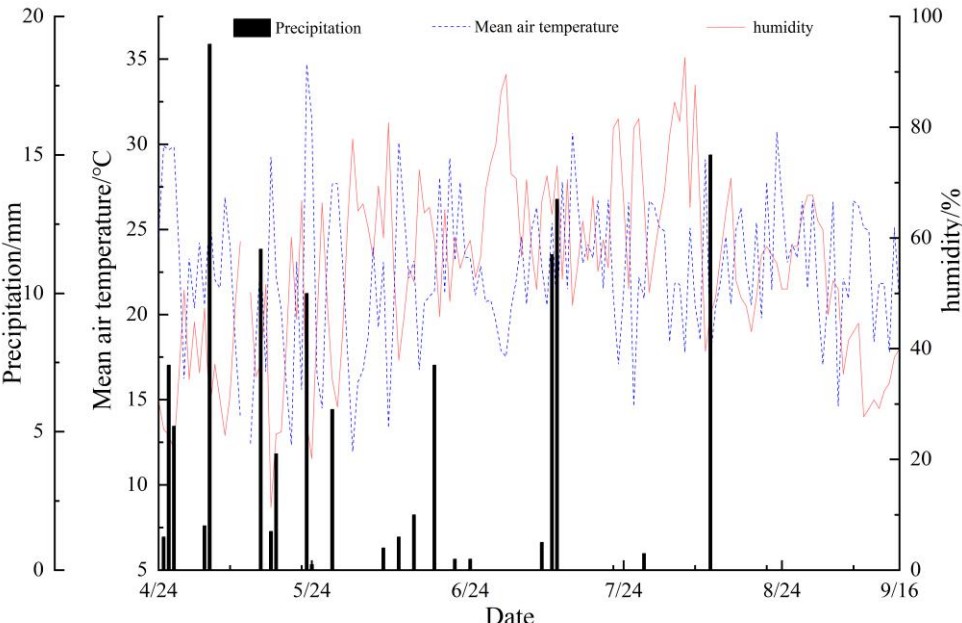

**Figure 2.** Daily precipitation (black bars), mean air temperature (blue) and humidity (red) from 24 April to 16 September for the study years (2019).

**Table 1.** Soil physical and chemical properties and contents of N, P and K at different depths in the experimental area.

| Soil Depth/(cm) | Grain Size Distribution /(mm) | Soil Type | Soil Bulk Density/ (g cm$^{-3}$) | Field Capacity/ (%) | The Organic Matter/ (g kg$^{-1}$) | N/(mg kg$^{-1}$) | P/(mg kg$^{-1}$) | K/(mg kg$^{-1}$) |
|---|---|---|---|---|---|---|---|---|
| 0–20 | 1.203 | Sandy soil | 1.55 | 30.89 | 15.16 | 120.02 | 45.21 | 203.98 |
| 20–40 | 0.985 | Sandy clay loam | 1.49 | 28.84 | 14.55 | 95.29 | 37.51 | 174.56 |
| 40–60 | 0.897 | Sandpaper clay loam | 1.51 | 25.12 | 11.49 | 82.38 | 32.12 | 197.38 |
| 60–80 | 1.157 | Sandy soil | 1.58 | 22.39 | 12.79 | 76.12 | 29.66 | 155.98 |
| 80–100 | 1.098 | Sandy soil | 1.59 | 20.76 | 9.42 | 64.09 | 22.78 | 176.97 |

## 2.2. Experimental Design

The experiment set two factors, namely irrigation amount and aerated irrigation; the irrigation volume was set to four levels: 4950 m$^3$ hm$^{-2}$ (W1), 4750 m$^3$ hm$^{-2}$ (W2), 4500 m$^3$ hm$^{-2}$ (W3), 4050 m$^3$ hm$^{-2}$ (W4). Aeration irrigation was set at two levels of aeration: 0% (A1) and 17% (A2). The experiment was performed with a two-factor completely randomized design. There are a total of eight treatments in the experiment, each treatment was set to repeat four times; there was a total of 32 plots, arranged in random blocks.

The total area of the experimental plots was 369 m$^2$, and each experimental plot was 15 m long and 6.15 m wide. A plastic film of 60 cm thickness was buried between each experimental district to eliminate the influence of water and salt infiltration between neighboring experimental districts. In each plot, processing tomatoes were sown under one strip of film (145 cm wide) with two drip capillaries, the bare land between mulches was about 30 cm (Figure 3). Figure 3 is the schematic diagram of the planting pattern. Processing tomato was planted in every mulching hole. The drip irrigation pipe spacing was 85 cm, and for the the single-wing labyrinth drip irrigation pipe (Xinjiang Tianye, Shihezi, China), the outer diameter was 16 mm and the wall thickness was 0.30 mm. The dripper spacing was 30 cm and the dripper flow rate was 1.8 L/h. The irrigation water in each experimental plot was controlled by a rotor flowmeter. Processing tomato seedlings were planted in a single hole with a single plant on both sides of thedrip tape, with row spacing of 35 cm and plant spacing of 35 cm. A Mazzei Air 1078 Injector (Mazzei Corp, Bakersfield, USA) was used for aeration treatment and installed at the head of the drip irrigation system. The pressure differential within the Mazzei Air Injector (inlet, 0.1 MPa; outlet, 0.02 MPa) was calibrated with pressure gauges on both sides and controlled by a pressure-regulated by pass tubule, and a volumetric air concentration of 17% was established in the aerated water [28,29]. There was a separate valve and water meter in each experimental plot.

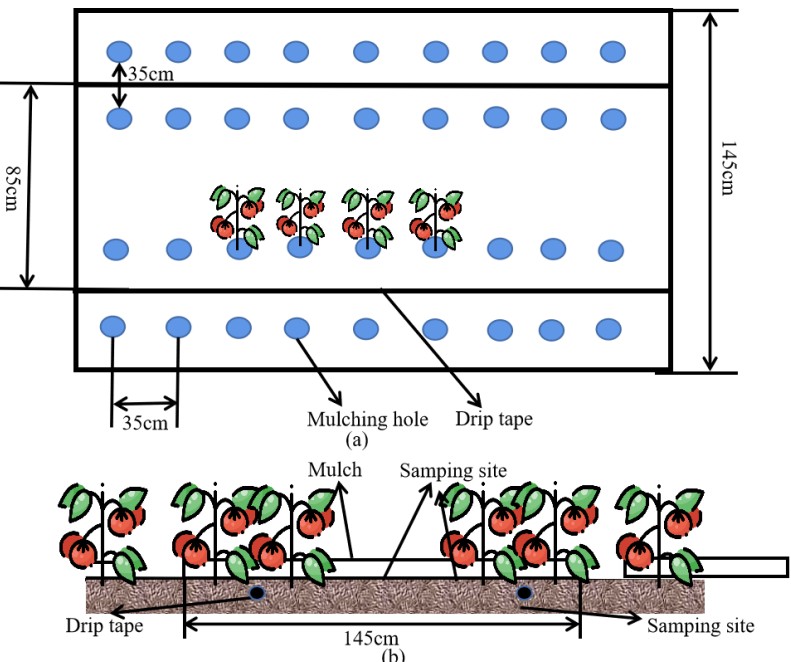

**Figure 3.** Vertical view (**a**) and front view (**b**) of processing tomato planting patterns.

The local variety "3166 Jinfan" was selected for the experiment. Seedlings were planted on 6 May, and 30 m$^3$ hm$^{-2}$ of water was irrigated before planting (not included in the irrigation practice) to ensure the survival of the processing tomato seedlings. After that, the water was irrigated for the first time after about 15 days, and then irrigated continuously eight times during the growing period (Table 2). The processing tomato was harvested on 22 August, and the total growth period was 109 days. Irrigation and fertilization

practice refer to the work of other scholars. The experimental fertilizers (Table 3) were urea $CO(NH_2)_2$ (N quality score was 46.4%), ammonium phosphate $NH_4H_2PO_4$ ($P_2O_5$ quality score was 60.5%) and potassium chloride KCl ($K_2O$ quality score was 57%), which were applied via drip irrigation laterals [30]. The management of other fields is the same as that of ordinary high-yield fields, and all agronomic managements, such as pruning and fertilization, followed local production practices.

**Table 2.** Irrigation schedule of processing tomato during the whole growth period.

| Treatment Group | Irrigation Amount/ $m^3 \ hm^{-2}$ | Irrigation Times | Irrigation Norm/ $m^3 \ hm^{-2}$ |
|---|---|---|---|
| A1W1 | 4950 $m^3 \ hm^{-2}$ | 8 | 618.25 |
| A1W2 | 4750 $m^3 \ hm^{-2}$ | 8 | 590.62 |
| A1W3 | 4500 $m^3 \ hm^{-2}$ | 8 | 562.50 |
| A1W4 | 4050 $m^3 \ hm^{-2}$ | 8 | 506.25 |
| A2W1 | 4950 $m^3 \ hm^{-2}$ | 8 | 618.25 |
| A2W2 | 4750 $m^3 \ hm^{-2}$ | 8 | 590.62 |
| A2W3 | 4500 $m^3 \ hm^{-2}$ | 8 | 562.50 |
| A2W4 | 4050 $m^3 \ hm^{-2}$ | 8 | 506.25 |

**Table 3.** Fertilizer schedule of processing tomato during the whole growth period.

| Growth Period | Date | Irrigation and Fertilization Cycle/d | Ratio of Irrigation and Fertilization/% | Frequency of Irrigations and Fertilizations |
|---|---|---|---|---|
| Seedling stage | 6 May–9 June | 35 | 16.7 | 1 |
| Florescence | 10 June–2 July | 23 | 16.7 | 2 |
| Fruit expansion | 3 July–4 August | 33 | 50 | 4 |
| Mature stage | 5 August–22 August | 18 | 16.6 | 1 |
| Whole growth period | 6 May–22 August | 109 | 100.0 | 8 |

*2.3. Measurement*

2.3.1. Soil Moisture Content

The soil samples were extracted from under the drip irrigation pipe and between the film with a twist drill in each growth period of the processing tomato. The sampling depth was 0~10 cm, 10~20 cm, 20~30 cm, 30~40 cm, 40~50 cm, 50~60 cm, 60~70 cm, 70~80 cm, 80~90 cm and 90~100 cm, and its fresh weight was recorded. They were then placed in an oven to dry and weighed dry. We then calculated the mass moisture content of the soil. The calculation formula is:

$$MMC = (FW - DW)/FW \tag{1}$$

where *MMC* is mass moisture content; *FW* is weight before drying; *DW* is weight after drying.

2.3.2. Growth Index of Processing Tomato
Plant Height

Plant height of three processing tomato plants was randomly selected for measurement at the seedling stage (18 May), flowering stage (12 June), fruit expansion stage (7 July) and mature stage (6 August). The plant height was recorded using a tape measure to measure the natural height from the root of the plant to the growth point of the main stem of the tomato in cm.

Stem Diameter

Stem diameter of three processing tomato plants was randomly selected for measurement at the seedling stage (18 May), flowering stage (12 June), fruit expansion stage (7 July)

and mature stage (6 August). The stem diameter was measured by a digital display vernier caliper at 3 cm above the root.

Leaf Area Index (LAI)

Leaf area index (LAI) of three processing tomato plants was randomly selected for measurement at the seedling stage (18 May), flowering stage (12 June), fruit expansion stage (7 July) and mature stage (6 August). The leaf area index of a single plant of processing tomato was calculated by using the leaf tracing method:

$$LAI = LA/(PLD \times ROWD) \tag{2}$$

where $LA$ is leaf area of processing tomato per plant; $PLD$ is the plant spacing (cm); $ROWD$ is the row spacing (cm).

Dry Matter

Dry matter of three processing tomato plants was randomly selected for measurement at the seedling stage (18 May), flowering stage (12 June), fruit expansion stage (7 July) and mature stage (6 August). The dry matter of aboveground parts (stems, leaves, flowers and fruits) was separately put into marked file bags. Fresh weight was recorded and then dry matter was placed in the oven. It was sterilized at 105 °C for 30 min, then dried at 75 °C to constant weight, and weighed after cooling.

2.3.3. Physiology Index of Processing Tomato

The photosynthetic characteristics of functional leaves (middle lobes of third pinnate compound leaves from top to bottom) of processing tomato were measured by a CI-340 hand-held photosynthetic apparatus (American LI-COR Corporation Lincoln, Lincoln, NE, USA), and the measured leaves were labeled. Three plants were measured continuously for each treatment. The photo-synthetic physiological indicators of processing tomato, such as net photosynthetic rate (Pn), transpiration rate (Tr), gas conductivity (Gs) and intercellular $CO_2$ concentration (Ci), were measured. Photosynthesis indexes of the processing tomatoes were measured at 12:00 on 5 July, 14 July, 26 July and 11 August, respectively. Diurnal variations of photo-synthesis indicators of processing tomatoes were measured at 2 h intervals over a 12 h period starting at 08:00 (seven times for one measurement day) in the 2019 experiment.

The fluorescence parameters were measured by a German Walz PAM2500 fluorometer (Walz, Nuremberg City, Germany). Fluorescence index of tomatoes during the processing period was measured on 28 June, 6 July, 12 July and 23 July. Diurnal variations of fluorescence changes of processing tomatoes were measured at 2:00, 8:00, 12:00, 16:00, 20:00 and 22:00 on 28 June. After shading the leaves for 20 min, the initial fluorescence yield $F_0$ and the maximum fluorescence yield $F_m$ were measured. The maximum photochemical efficiency $F_v/F_m$ and potential photochemical activity $F_v/F_0$ of PSII were calculated according to the formula.

Fluorescence index was calculated as followed:

$$F_v/F_m = (F_m - F_0)/F_m \tag{3}$$

$$F_v/F_0 = (F_m - F_0)/F_0 \tag{4}$$

where $F_0$ is the initial fluorescence yield, $F_v$ is the variable fluorescence yield and $F_m$ is the maximum fluorescence yield.

2.3.4. Yield and Irrigation Water Use Efficiency

Yield data (including fruit yield per plant, fruit weight and number per plant) were recorded for fruit harvested from five plants from the middle of the plot. The BWS-SN-30 electronic weighing table scale (Xiamen Berens brand, Xiamen, China) was used to record yield per plant.

Irrigation water use efficiency (*IWUE*) was calculated as follows:

$$IWUE = Y/I \tag{5}$$

where *Y* is fruit yield per plant (g); *I* is the irrigation amount (L).

2.3.5. Fruit Quality Index of Processing Tomato

The quality of vitamin C, soluble solids, total sugars and total acids in the processing tomatoes was determined at harvest time (22 August). We measured from five mature fruits at harvest for each experimental plot. Soluble solids were determined by a handheld refractometer (MAST-3M, Sao Paulo, Japan). The content of vitamin C was determined by titration. The content of total sugar was determined by enthrone colorimetry. The content of total acid was determined by the acid-base titration indicator method. The sugars–acids ratio is equal to total sugars divided by total acids.

*2.4. Data Analysis*

2016 Excel was used for experimental data processing and calculation and Origin9.0 was used for drawing. Variance analysis, significance analysis, relevance analysis and Principal Component Analysis (PCA) were performed using SPSS 20.0 (IBM SPSS Statistics, New York, NY, USA). The different treatments were analyzed by one-way analysis of variance (ANOVA). The different factors were analyzed by two-way analysis of variance (ANOVA). The least significant difference was determined when ANOVA indicated significant differences ($p < 0.05$). All statistical analyses were conducted to the $p < 0.05$ level, unless stated otherwise. Fisher's protected least significant difference (LSD) test at the 0.05 or 0.10 significance level was used to compare differences between treatment means.

**3. Results**

*3.1. Soil Moisture Content*

The average weighted moisture content of soil profiles treated with different irrigation level and aerated irrigation in the fruit expansion stage is shown in Figure 4. The results could reflect the distribution characteristics of soil water content in the direction of a 0~70 cm vertical drip irrigation area with a 0~40 cm horizontal drip irrigation area under the drip tape. In terms of irrigation quota level, the average water content of the 0~70 cm depth soil layer of W1, W2 and W3 under drip tape increased by 31.5%, 22.3% and 14.2%, compared with that of W4. Compared with W4, the average water content of the 0~70 cm depth soil layer 20 cm away from the drip tape in W1, W2 and W3 increased by 39.8%, 30.8% and 18.5%. In the average water content of the 0~70 cm depth soil layer 40 cm away from the drip tape, W4 was increased by 28.0%, 20.2% and 8.4%, compared to W1, W2 and W3. The soil moisture content increased with the increase in irrigation quota. The change in range of soil moisture content caused by the irrigation quota first increased and then decreased with the increase of distance from the drip irrigation belt. From the perspective of aeration level, the average soil moisture content of 0~70 cm soil layer under the A1 drip irrigation belt increased by 15.5% compared with A2. The average soil moisture content of the 0~70 cm depth soil layer at 20 cm from A1 to A2 had no significant change. The average soil moisture content of the 0~70 cm depth soil layer at 20 cm away from the drip irrigation belt of A1 was 20% lower than that of A2. Aeration treatment can reduce the soil moisture content of 0~70 cm depth under drip irrigation. When the distance from the drip tape is longer, the aerated treatment will increase the soil moisture content.

*3.2. Stem Diameter, Plant Height, Leaf Area Index and Dry Matter of Processing Tomato*

Figure 5a–c analyzed the effects of water-air coupling on stem diameter (a), plant height (b) and leaf area index (c) of processing tomatoes. The stem diameter of W2, W3 and W4 increased by 10.8%, 10.4% and 4.2% compared with W1 at seedling stage. At florescence, the stem diameter W1, W2 and W3 increased by 2.7%, 8.8% and 14.5% compared with W4. The stem diameter of W2, W3 and W4 increased by 14.4%, 10.9% and 3.7% compared with

W1 at fruit expansion. At the mature stage, the stem diameter of W1, W2 and W3 increased by 1.9%, 11.4% and 7.2% compared with W4. The stem diameter increased first and then decreased with the increase in irrigation quota. In terms of aeration level, A2 increased by 8.9% compared with A1 at the seedling stage. At florescence, A2 increased by 8.9% compared with A1. The stem diameter of A2 increased by 7.3% compared with A1 at fruit expansion. At the mature stage, A2 increased by 9.6% compared with A1.

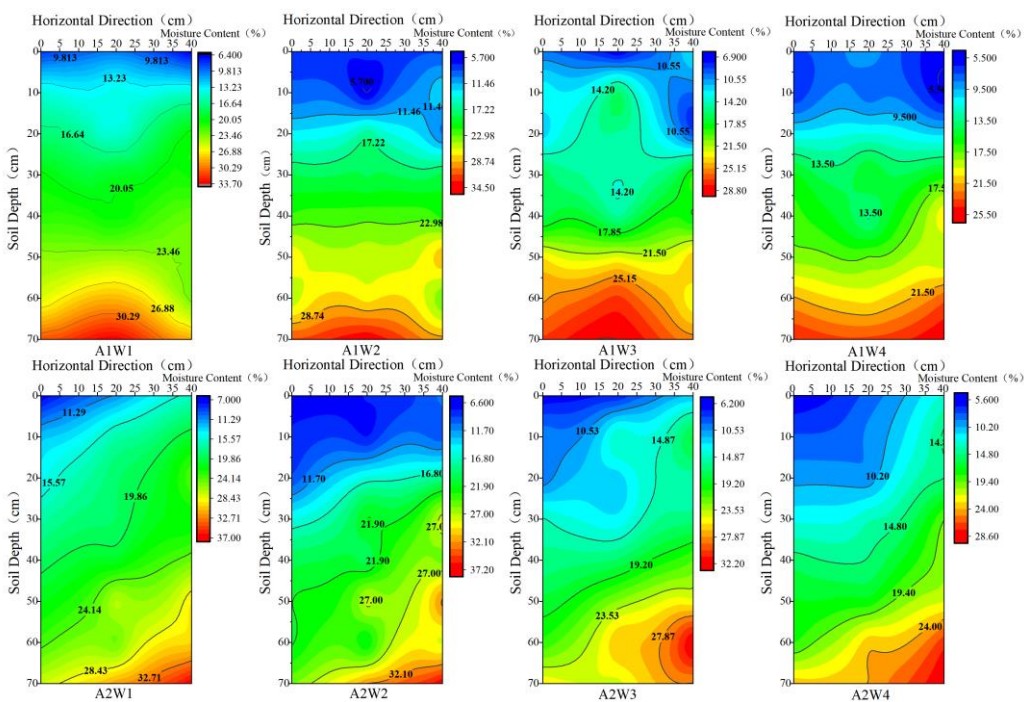

**Figure 4.** Soil moisture content of processing tomato in horizontal direction of 0~40 cm and vertical direction of 0~70 cm measured on 14 July (two days after irrigation) for A1 and A2 treatments under W1, W2, W3 and W4 irrigation levels.

From the irrigation quota level, the plant height of W1, W2 and W3 increased by 7.1%, 9.3% and 10.7% compared with W4 at the seedling stage. At florescence, the plant height of W1, W2 and W3 increased by 2.7%, 3.9% and 3.9% compared with W4. The plant height of W1, W2 and W3 increased by 10.7%, 12.6% and 12.1% compared with W4 at fruit expansion. At the mature stage, the plant height of W1, W2 and W3 increased by 3.3%, 5.5% and 5.7% compared with W4. The plant height increased most obviously in the fruit expansion stage. In terms of aeration level, A2 increased by 3.3% compared with A1 at the seedling stage. At florescence, A2 increased by 9.6% compared with A1. The plant height of A2 increased by 10.2% compared with A1 at the fruit expansion stage. At the mature stage, A2 increased by 8.4% compared with A1. Aerated treatment promoted plant height growth.

From the irrigation quota level, the leaf area index (LAI) of W1, W2 and W3 increased by 4.9%, 48.2% and 40.5% compared with W4 at the seedling stage. At florescence, the LAI of W1, W2 and W3 increased by 14.5%, 32.1% and 25.5% compared with W4. The LAI of W1, W2 and W3 increased by 6.7%, 19.8% and 17.4% compared with W4 at fruit expansion. At the mature stage, the LAI of W1, W2 and W3 increased by 6.7%, 26.2% and 21.5% compared with W4. The LAI increased with the increase in irrigation quota. In terms of aeration level, A2 increased by 28.9% compared with A1 at the seedling stage. At florescence, A2 increased by 25.5% compared with A1. The LAI of A2 increased by 6.2% compared with A1 at the fruit expansion stage. At the mature stage, A2 increased by 10.9% compared with A1.

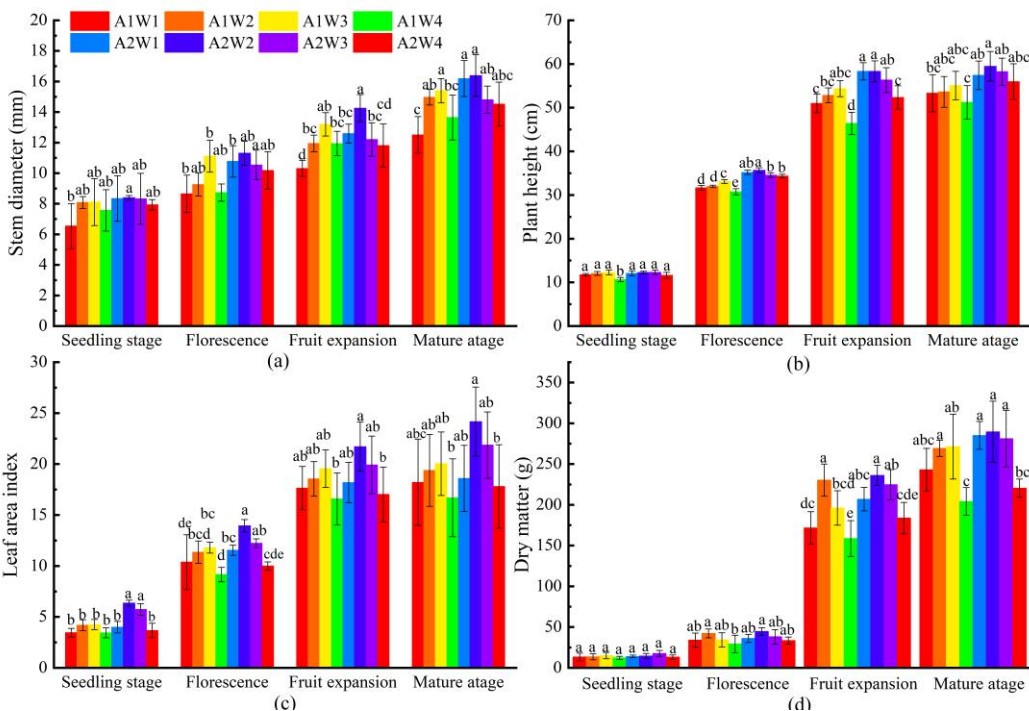

**Figure 5.** Stem diameter (**a**), plant height (**b**), leaf area index (**c**) and biomass (**d**) of processing tomato in seeding stage, florescence, fruit expansion and mature stage for eight treatments. Different letters in the same column indicate significance at $p < 0.05$. The error bars show $\pm$ standard deviation and the n value is 3 (replicated three times).

Figure 5d showed that aerated irrigation had a significant effect on dry matter in each growing period of processing tomato. From the irrigation quota level, the dry matter of W1, W2 and W3 increased by 8.2%, 11.0% and 28.8% compared with W4 at the seedling stage. At florescence, the dry matter of W1, W2 and W3 increased by 11.7%, 38.3% and 15.2% compared with W4. The dry matter of W1, W2 and W3 increased by 10.6%, 36.2% and 22.9% compared with W4 at fruit expansion. At the mature stage, the dry matter of W1, W2 and W3 increased by 24.4%, 31.6% and 30.1% compared with W4. The dry matter of W2 was higher than other levels throughout the whole growth period. In terms of aeration level, A2 increased by 9.3% compared with A1 at the seedling stage. At florescence, A2 increased by 8.8% compared with A1. The dry matter of A2 increased by 12.5% compared with A1 at the fruit expansion stage. At the mature stage, A2 increased by 9.0% compared with A1. The dry matter of processing tomato changed the fastest during fruit expansion.

*3.3. Photosynthetic Characteristics of Processing Tomato*

Figure 6 shows the effects of aeration treatment on the photosynthetic characteristics of processing tomato. On 12 July, the net photosynthetic rate (Pn) of W2 treatment was the highest, which was significantly different from that of other treatments ($p < 0.05$). On 26 July, the net photosynthetic rate of W2 treatment was the lowest, which was significantly different from that of other treatments ($p < 0.05$). On 12 July and 26 and 11 August, the net photosynthetic rate of W1 treatment was the highest and significantly different from that of other treatments ($p < 0.05$). Stomatal conductance (Gs) of A2W1 and A2W3 showed a trend of decreasing first and then increasing, while those of A2W2 and A2W4 showed a trend of increasing first and then decreasing. The stomatal conductance of treatment A2W1 was the highest on 5 July, which was significantly different from that of treatments A2W2, A2W3 and A2W4 ($p < 0.05$). A2W1 treatment had the highest stomatal conductance on 26 July, and there was significant difference among all treatments ($p < 0.05$). A2W3 treatment had the highest stomatal conductance on 11 August, and there was significant difference among all treatments ($p < 0.05$). The transpiration rate (Tr) showed a trend of decreasing first and then increasing. The intercellular $CO_2$ concentration (Ci) showed a

trend of decreasing first and then increasing. The transpiration rate of A2W3 treatment was the highest on 12 July and 11 August, and it was significantly different from that of the other treatments ($p < 0.05$).

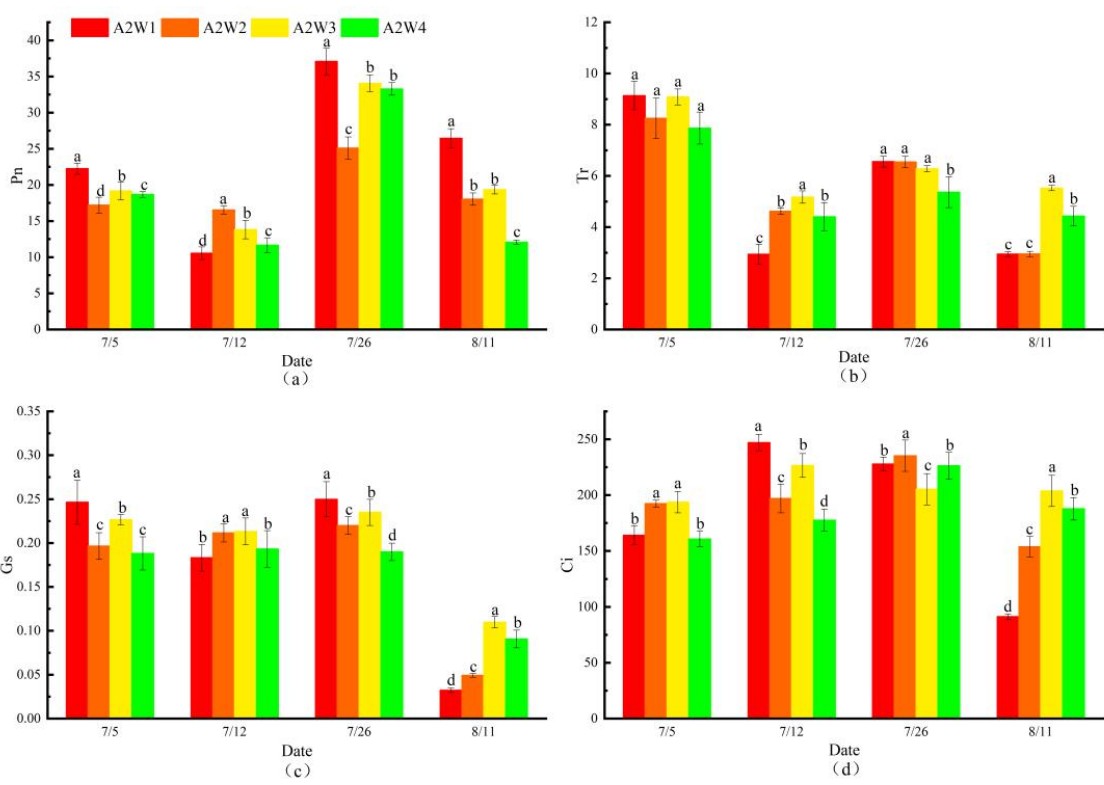

**Figure 6.** The variation photosynthetic index which includes Pn—net photosynthetic rate (**a**), Tr—transpiration rate (**b**), Gs—stomatal conductance (**c**) and Ci—intercellular $CO_2$ concentration (**d**) for aeration treatments at four (W1, W2, W3, and W4) irrigation levels. Different letters in the same column indicate significance at $p < 0.05$. The error bars show $\pm$ standard deviation and the n value is 3 (replicated three times).

Figure 7 represents the analysis of diurnal variation of photosynthetic indexes. Under the aeration condition, the diurnal variation curve of net photosynthetic rate of processing tomato leaves under drip irrigation at different irrigation levels showed a trend of increasing firstly and then decreasing; the peak occurs at 12:00. A2W2 treatment was significantly different from other treatments. In terms of Pn, A2W1 treatment was the highest. The variation curve of stomatal conductance of processing tomato leaves treated with different irrigation levels was similar to that of the net photosynthetic rate, which showed a trend of increasing first and then decreasing; the peak came at 14:00. The performance of A2W1 treatment was the most obvious, which was significantly higher than other treatments. The diurnal variation of intercellular $CO_2$ concentration (Ci) of processing tomato is a fluctuating variation. The trough value appeared at 12 o'clock, and A2W2 treatment was higher than other treatments. In daily variation, stomatal conductance is a fluctuating variation.

Figure 8 shows the effects of aeration treatments at different times under different irrigation levels on the fluorescence characteristics of processing tomatoes. On 28 June, the $F_v/F_0$ of A2W4 was the highest, which was significantly different from other treatments ($p < 0.05$). The trend of $F_v/F_m$ was similar to that of $F_v/F_0$. On 6 July, there was no significant difference between $F_v/F_0$ and $F_v/F_m$. On 12 July, the $F_v/F_0$ of A2W1 was the lowest, which was significantly different from that of other treatments. On 23 July, there was no significant difference between $F_v/F_0$ and $F_v/F_m$.

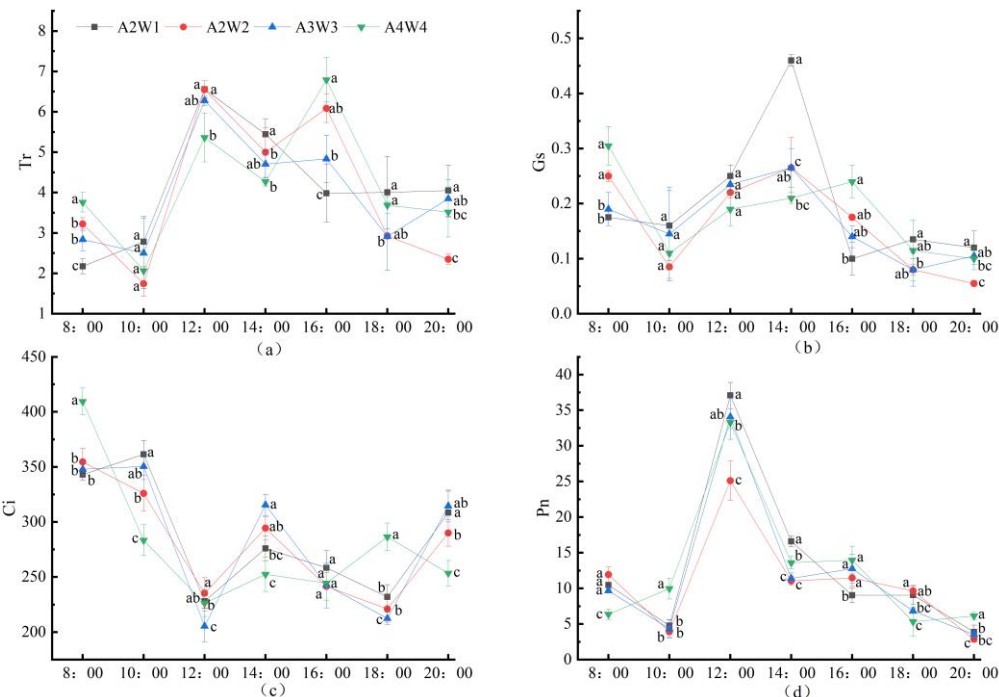

**Figure 7.** The variation photosynthetic index which includes Pn—net photosynthetic rate (**a**), Tr—transpiration rate (**b**), Gs—stomatal conductance (**c**) and Ci—intercellular $CO_2$ concentration (**d**) for aeration treatments at different times on 14 July. Different letters in the same column indicate significance at $p < 0.05$. The error bars show ± standard deviation and the n value is 3 (replicated three times).

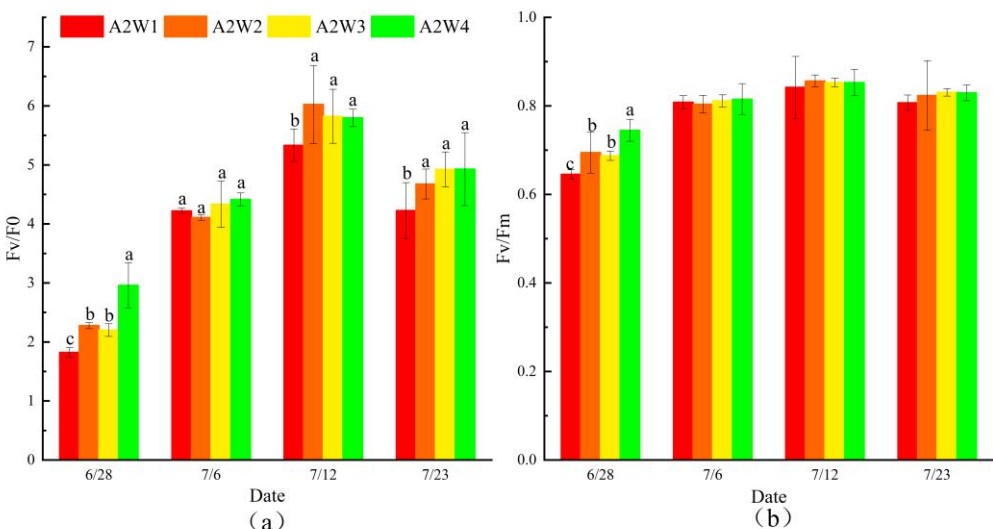

**Figure 8.** Fluorescence changes in aerated group, the horizontal axis shows the test date. Test indicators include $F_v/F_0$ (**a**) and $F_v/F_m$ (**b**). $F_0$ is the initial fluorescence yield, $F_v$ is the variable fluorescence yield and $Fm$ is the maximum fluorescence yield. Different letters in the same column indicate significance at $p < 0.05$. The error bars show ± standard deviation and the n value is 3 (replicated three times).

Figure 9 shows the effect of aeration treatment on the daily variation of fluorescence characteristics of processing tomato under different irrigation levels. It can be seen from Figure 9a,b that the diurnal variation trend of $F_v/F_0$ and $F_v/F_m$ in processed tomato leaves is consistent. It shows a trend of decreasing first and then increasing, and the valley values all appeared at 16:00.

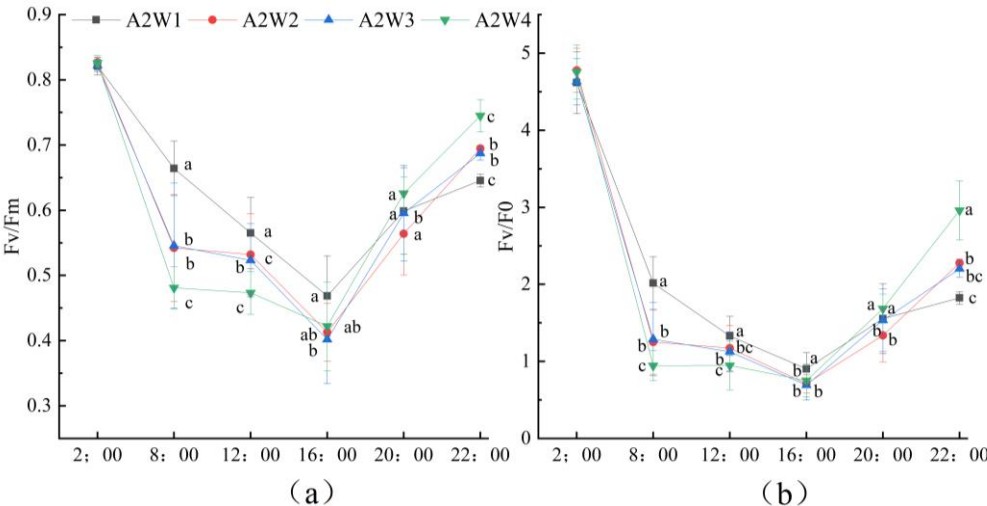

**Figure 9.** Diurnal changes in fluorescence characteristics of aerated group on 28 June, and the horizontal axis shows the test time. Test indicators include $F_v/F_0$ (**a**) and $F_v/F_m$ (**b**). $F_0$ is the initial fluorescence yield, $F_v$ is the variable fluorescence yield and $Fm$ is the maximum fluorescence yield. Different letters in the same column indicate significance at $p < 0.05$. The error bars show ± standard deviation and the n value is 3 (replicated three times).

### 3.4. Yield Factors of Processing Tomato Fruits

The effects of different irrigation water and air combination schemes on yield per plant, single fruit weight, number of fruits per plant and irrigation water use efficiency of drip irrigation in processing tomatoes are shown in Table 4. The F values showed that both A and W had high significant ($p < 0.01$) effects on yield per plant. A1W1 was significantly increased by 28.15% compared with A1W4. A2W1 was significantly increased by 7.56% and 10.19% compared with that at W3 and W4 levels, respectively. The F values showed that only factor A had high significant ($p < 0.01$) effects on single fruit weight. Under the same irrigation amount, the single fruit weight of A2 was higher than that of A1. Compared with A1W4, A2W4 increased by 14.12% at most. The F values showed that only factor W had high significant ($p < 0.01$) effects on the number of fruits per plant. Under A1 and A2 conditions, number of fruits per plant increased with the increase of irrigation amount. The maximum number of fruits per plant of A1W1 was 60.67. The F values showed that only factor A had high significant ($p < 0.01$) effects on irrigation water use efficiency. Under the same irrigation amount, the irrigation water use efficiency of A2 was higher than that of A1. Compared with A1W1, A2W1 increased by 10.31% at most.

**Table 4.** Effects of different treatment methods on the yield factors (yield per plant, single fruit weight, number of fruits per plant, irrigation water use efficiency) of processing tomato fruits.

| Treatment Group | Yield per Plant | Single Fruit Weight | Number of Fruits per Plant | Irrigation Water Use Efficiency |
|---|---|---|---|---|
| A1W1 | 3.87 ± 0.25 abc | 60.25 ± 3.40 abc | 60.67 ± 1.15 a | 34.71 ± 2.38 a |
| A1W2 | 3.49 ± 0.40 bc | 61.30 ± 7.66 abc | 57.00 ± 2.00 bc | 34.53 ± 3.96 a |
| A1W3 | 3.27 ± 0.11 cd | 59.74 ± 5.36 bc | 55.00 ± 3.61 cd | 34.18 ± 1.15 a |
| A1W4 | 3.02 ± 0.07 d | 55.82 ± 2.07 c | 53.00 ± 1.73 de | 34.31 ± 0.82 a |
| A2W1 | 4.03 ± 0.13 a | 68.36 ± 1.39 ab | 59.00 ± 1.00 ab | 38.29 ± 1.19 a |
| A2W2 | 3.84 ± 0.18 ab | 69.36 ± 3.78 a | 55.33 ± 0.58 cd | 37.96 ± 1.74 a |
| A2W3 | 3.57 ± 0.29 bc | 66.55 ± 6.79 ab | 53.67 ± 1.15 cde | 37.25 ± 3.07 a |
| A2W4 | 3.24 ± 0.14 cd | 63.70 ± 5.39 abc | 51.00 ± 2.65 e | 37.59 ± 1.64a |
| F value test | | | | |
| W | 12.744 ** | 1.443 | 16.932 ** | 0.142 |
| A | 12.891 ** | 14.676 ** | 4.301 | 13.412 ** |
| W × A | 0.06 | 0.023 | 0.029 | 0.014 |

The different letters in the same column indicate significant differences at the level of $p = 0.05$, ** indicate that there is a significant difference at $p = 0.01$ levels.

### 3.5. Fruit Quality Factors of Processing Tomato

The effects of different irrigation water and air combination schemes on vitamin C, soluble solids, soluble sugar, organic acid and sugar–acid ratio of drip irrigation in processing tomatoes are shown in Table 5. The F values showed that both A and W had significant ($p < 0.05$) effects on vitamin C, soluble solids, soluble sugar and organic acid. As for vitamin C, W2 was 17.0%, 3.3% and 10.2% significantly higher than W1, W3 and W4, respectively; A2 was 4.0% significantly higher than A1. As for Soluble solids, W2 was 23.3%, 6.3% and 15.7% significantly higher than W1, W3 and W4, respectively; A2 was 4.4% significantly higher than A1. As for soluble sugar, W4 was 14.7%, 6.4% and 4.0% significantly higher than W1, W2 and W3, respectively. The smaller the irrigation quota, the higher the content of soluble sugar. A2 was 3.2% significantly higher than A1. As for Organic acid, W4 was 18.4%, 11.5% and 5.5% significantly higher than W1, W2 and W3, respectively. A2 was 3.8% significantly higher than A1.

**Table 5.** Effects of different treatment methods on fruit quality factors (vitamin C, soluble solids, soluble sugar, organic acid, sugar-acid ratio) of processing tomato.

| Treatment Group | Vitamin C/mg 100 g$^{-1}$ | Soluble Solids/% | Soluble Sugar/% | Organic Acid/% | Sugar-Acid Ratio |
|---|---|---|---|---|---|
| A1W1 | 15.87 ± 0.15 e | 4.75 ± 0.23 e | 5.11 ± 0.11 g | 0.24 ± 0.00 f | 21.46 ± 0.40 a |
| A1W2 | 18.35 ± 0.67 abc | 5.88 ± 0.23 ab | 5.49 ± 0.06 e | 0.25 ± 0.00 e | 21.69 ± 0.33 a |
| A1W3 | 17.77 ± 0.46 bc | 5.55 ± 0.08 bc | 5.64 ± 0.05 d | 0.27 ± 0.00 d | 21.13 ± 0.28 ab |
| A1W4 | 16.54 ± 0.59 de | 5.04 ± 0.15 de | 5.88 ± 0.11 b | 0.29 ± 0.00 b | 20.61 ± 0.06 b |
| A2W1 | 16.17 ± 0.34 e | 4.98 ± 0.22 de | 5.28 ± 0.02 f | 0.25 ± 0.01 e | 21.00 ± 0.47 ab |
| A2W2 | 19.13 ± 0.92 a | 6.12 ± 0.31 a | 5.70 ± 0.05 cd | 0.27 ± 0.00 d | 21.43 ± 0.14 a |
| A2W3 | 18.51 ± 0.44 ab | 5.73 ± 0.26 b | 5.82 ± 0.05 bc | 0.28 ± 0.00 c | 21.11 ± 0.35 ab |
| A2W4 | 17.47 ± 0.56 cd | 5.33 ± 0.05 cd | 6.03 ± 0.13 a | 0.29 ± 0.00 a | 20.45 ± 0.72 b |
| F-value | | | | | |
| W | 28.245 ** | 34.328 ** | 99.316 ** | 147.996 ** | 7.19 |
| A | 9.068 ** | 7.471 * | 31.26 ** | 50.748 ** | 1.931 |
| W×A | 0.353 | 0.068 | 0.228 | 0.572 | 0.336 |

The different letters at same column indicate significant differences at the level of $p = 0.05$. * and **, respectively, indicate that there is a significant difference at $p = 0.05$ and $p = 0.01$ levels. 2. * is the interaction effect of experiment factors of irrigation level (W) and method (AI).

Table 6 showed that there are seven pairs of significant or extremely significant correlations among the nine indexes of processing tomato yield and quality. In yield index, yield per plant was significantly positively correlated with fruit weight per plant, while single fruit weight was significantly positively correlated with IWUE. In addition, the number of fruits per plant was negatively correlated with soluble sugar and organic acids. Among the quality indexes, vitamin C was significantly positively correlated with soluble solids, soluble sugars were significantly positively correlated with organic acids and organic acids were significantly negatively correlated with sugar–acid ratio.

Through principal component analysis, the three main components were extracted based on the eigenvalues greater than 1, and the cumulative variance contribution rate was 99.274% (Table 7). The variance contribution rate of the first principal component is 46.750%, which mainly positively affected the yield per plant, number of fruits per plant and the sugar acid ratio. It can be seen in Table 8 that their contribution rate is fruit number per plant > per plant yield > sugar–acid ratio, and which mainly negatively affected soluble sugar and organic acid. The variance contribution rate of the second principal component was 33.660%, which was mainly affected by single fruit weight, IWUE, vitamin C and soluble solids. It can be seen from Table 8 that the contribution rate is the utilization efficiency of single fruit weight > soluble solid > vitamin C > IWUE. The variance contribution rate of the third principal component was 18.864%, which was mainly affected by the positive effects of the sugar–acid ratio, soluble solids and vitamin C, and the negative effects of IWUE. The contribution rate of IWUE was higher than that of the sugar–acid ratio, soluble solids and vitamin C. Comprehensive correlation showed that

IWUE was negatively correlated with sugar–acid ratio, soluble solids and vitamin C. With the increase of IWUE, sugar–acid ratio, soluble solids and vitamin C decreased.

**Table 6.** Correlations among various fruit yield factors (yield per plant, single fruit weight, number of fruits per plant, irrigation water use efficiency) and fruit quality factors (vitamin C, soluble solids, soluble sugar, organic acid, sugar-acid ratio).

| Index | Yield per Plant | Single Fruit Weight | Number of Fruits per Plant | Irrigation Water Use Efficiency | Vitamin C | Soluble Solids | Soluble Sugar | Organic Acid | Sugar-Acid Ratio |
|---|---|---|---|---|---|---|---|---|---|
| Yield per plant | 1.000 | 0.813 * | 0.661 | 0.611 | 0.041 | 0.097 | −0.626 | −0.656 | 0.523 |
| Single fruit weight | | 1.000 | 0.102 | 0.908 ** | 0.407 | 0.411 | −0.061 | −0.131 | 0.183 |
| Number of fruits per plant | | | 1.000 | −0.137 | −0.435 | −0.346 | −0.995 ** | −0.966 ** | 0.681 |
| Irrigation water use efficiency | | | | 1.000 | 0.224 | 0.197 | 0.166 | 0.144 | −0.217 |
| Vitamin C | | | | | 1.000 | 0.988 ** | 0.485 | 0.305 | 0.312 |
| Soluble solids | | | | | | 1.000 | 0.399 | 0.210 | 0.391 |
| Soluble sugar | | | | | | | 1.000 | 0.966 ** | −0.641 |
| Organic acid | | | | | | | | 1.000 | −0.761 * |
| Sugar−acid ratio | | | | | | | | | 1.000 |

The different letters at same column indicate significant differences at the level of $p = 0.05$. * and **, respectively, indicate that there is a significant difference at $p = 0.05$ and $p = 0.01$ levels.

**Table 7.** Eigenvalue and variance contribution rates based on the principal component analysis.

| Principle Components | Eigenvalue | Variance Contribution Rate/% | Cumulative Variance Contribution Rate/% |
|---|---|---|---|
| 1 | 4.207 | 46.750 | 46.750 |
| 2 | 3.029 | 33.660 | 80.409 |
| 3 | 1.698 | 18.864 | 99.274 |
| 4 | 0.036 | 0.394 | 99.668 |
| 5 | 0.020 | 0.227 | 99.895 |
| 6 | 0.009 | 0.101 | 99.996 |
| 7 | 0.000 | 0.004 | 100.000 |
| 8 | $1.381 \times 10^{-16}$ | $1.535 \times 10^{-15}$ | 100.000 |
| 9 | $-2.669 \times 10^{-16}$ | $-2.965 \times 10^{-15}$ | 100.000 |

**Table 8.** The component matrix based on the rotary factor method.

| Index | Principle Components | | |
|---|---|---|---|
| | 1 | 2 | 3 |
| Yield per plant | 0.756 | 0.582 | −0.292 |
| Single fruit weight | 0.241 | 0.886 | −0.392 |
| Number of fruits per plant | 0.987 | −0.139 | 0.032 |
| Irrigation water use efficiency | −0.012 | 0.732 | −0.677 |
| Vc | −0.347 | 0.781 | 0.518 |
| Soluble solids | −0.260 | 0.783 | 0.558 |
| Soluble sugar | −0.979 | 0.195 | −0.011 |
| Organic acid | −0.976 | 0.054 | −0.145 |
| Sugar-acid ratio | 0.709 | 0.294 | 0.632 |

Combined with the variance contribution rates of the three principal components, the linear function of comprehensive evaluation based on yield and quality of each treatment was obtained:

$$Z = 0.46750Z1 + 0.33660Z2 + 0.18864Z3 \tag{6}$$

where $Z$ is the comprehensive score of each treatment based on yield and fruit quality; $Z1$ is the comprehensive score of main factor 1; $Z2$ is the comprehensive score of main factor 2; $Z3$ is the comprehensive score of main factor 3. After standardization, the original data of

each yield and quality index can be replaced by Formula (6) to obtain the comprehensive score and comprehensive ranking of each treatment (Table 8). First of all, aeration irrigation treatment (A2W2) ranked first in the overall ranking (Table 9), and the Z1, Z2 and Z3 values of this treatment were all positive.

**Table 9.** The comprehensive score of the eight treatments, calculated by principal component analysis based on the impacts of these treatments on fruit yield index, IWUE and fruit quality index.

| Treatment | Main 1 Factor Z1 | Main 2 Factor Z2 | Main 3 Factor Z3 | Comprehensive Score Z | Comprehensive Ranking |
|---|---|---|---|---|---|
| A1W1 | 3.021389341 | −1.942167085 | −0.042102298 | 0.756314743 | 4 |
| A1W2 | 1.015580301 | 0.301942387 | 2.059624611 | 0.972001919 | 2 |
| A1W3 | −0.573979059 | −0.776880757 | 1.136918415 | −0.317671277 | 6 |
| A1W4 | −2.285599218 | −2.453827944 | −0.049594969 | −1.917754614 | 8 |
| A2W1 | 2.365940698 | 0.387204693 | −2.120478134 | 0.842520076 | 3 |
| A2W2 | 0.119291817 | 2.873421523 | 0.379559188 | 1.102567293 | 1 |
| A2W3 | −0.989880052 | 1.516464695 | −0.058859821 | 0.036837213 | 5 |
| A2W4 | −2.672743827 | 0.093825085 | −1.305053961 | −1.474818779 | 7 |

## 4. Discussion

Aerated drip irrigation is helpful in enhancing aerobic respiration, weakening the anaerobic respiration process, improving root respiration efficiency and facilitating the absorption of mineral nutrients [31]. Plant root growth is closely related to the growth and development of the aboveground part, and the two promote and influence each other [32,33]. Bagatur [34] thinks that aerated irrigation is good for onion growth. Lu et al. [35] found that the plant height and stem diameter increased by 13.90% and 4.13% with the same amount of irrigation by aerated irrigation. Wen et al. [36] found that the plant height and stem diameter of tomatoes under aerated irrigation increased by 1.44% and 3.02% compared with those under non-aerated irrigation. This study showed that the soil moisture of aerated irrigation was higher than that of conventional irrigation. Then, aerated irrigation could promote the growth of processed tomato. For example, the plant height and stem diameter of processed tomato under aerated irrigation increased by 10.2% and 7.3% compared with that under non-aerated irrigation. In addition, aerated irrigation promoted crop growth more obviously at florescence.

Reactive oxygen species (ROS) affect abscisic acid (ABA) signal transduction [37]. A plant body type of ABA can improve the water retention of the leaf, the root of the ABA with transpiration stream spread to the ground [38,39]. ABA can induce bud dormancy, leaf abscission and inhibit cell growth, and leaves respond quickly to ABA [40,41]. The photosynthetic reaction rate was affected by ABA. In addition, the chloroplast structure will change under stress, which will affect the membrane system, resulting in damage to the light and reaction-related enzyme system, and directly affect the photosynthetic intensity of crops [42–44]. Photosynthesis is the basis of crop growth and yield formation, and the high photosynthetic rate can improve crop yield [45]. Chen et al. [46] found that aeration treatment can effectively relieve hypoxia stress in the root zone. Under aeration treatment, the photosynthesis of leaves was enhanced, which resulted in the increase in dry matter mass of tomatoes [25]. In this study, the dry matter mass of processed tomato with aerated irrigation was higher than that non-aerated irrigation. The dry matter weight increased the most at fruit expansion.

Crop yield is affected by light, temperature, moisture, fertilizer and other factors [47,48]. Aerated drip irrigation can improve fruit yield and quality [25,49]. Chen et al. [50] studied the effect of aerated drip irrigation on the yield and quality of tomatoes in greenhouse. It was found that the fruit quality of tomato per plant after the root zone was aerated was higher than that under the same condition without aeration. Furthermore, Jia et al. [51] found that the study of aerated irrigation can significantly increase the yield of watermelon. Goorahoo et al. and Gadissa et al. [52,53] found that the environmental conditions of silty

clayey loam and clayey loam could be improved obviously by using aerated drip irrigation, and the yield of pepper could be likewise increased. In this paper, the results of processing tomato in field with aerated irrigation are consistent with the research. Among them, A2W2 treatment output increased by 31.3% compared with A1W2.

The nutritional quality (soluble sugars, organic acids, vitamin C, soluble solids) of crops has been paid more and more attention by researchers and consumers. Its content determines the nutritional value and taste of crops, and then affects the commercial value of crops [45,54]. Li et al. [55] found that aeration treatments increased vitamin C and sugar–acid ratio by 2%, and 43%, respectively, with the no aeration. In this test, with the increase in irrigation quota, vitamin C and soluble solids showed a trend of increasing first and then decreasing. Vitamin C and soluble solids in aerated treatment were increased by 14.0% and 4.4% compared with those in non-aerated treatment. Previous studies have also shown that appropriate reduction in soil water content can increase the content of vitamin C in tomato fruits [56]. That is to say, the increase in irrigation amount may lead to the increase in fruit water content, so that the content of quality indicators will be reduced due to dilution; AI can improve the situation.

The simultaneous improvement of yield and quality has become the main goal pursued by water-saving agriculture nowadays. However, generally speaking, the improvement in quality is often accompanied by a decline in yield [57]. The paper showed that aeration treatment could promote an increase in crop yield and quality under the same irrigation quota. According to the comprehensive analysis, the A2W2 treatment is the best for the growth, fruit yield and quality of processed tomato, which also provides a certain technical reference for the irrigation method of processed tomato to a certain extent.

## 5. Conclusions

Under different irrigation quota conditions, the growth rates of plant height, stem diameter and leaf area per plant under aerated irrigation were significantly increased, which showed obvious growth advantages at the seedling stage, flowering stage and fruit-setting stage, and similar growth rates at later growth stages.

Under aerated irrigation, the cumulative yield of tomato increased by 9.7% on average compared with conventional irrigation, and the yield per plant under A2W1, A2W2, A2W3 and A2W4 treatment increased by 10.3%, 9.9%, 8.9% and 9.5%, respectively, compared with conventional irrigation. The characteristics of tomato yield distribution were that the fruit ripened earlier, granting the potential to obtain greater economic benefits.

Aerated irrigation was superior to conventional irrigation in tomato quality, and the average value of vitamin C content and soluble solid content under A2W1, A2W2, A2W3 and A2W4 treatment increased by 4.0% and 4.4%, respectively, suggesting that aerated irrigation could improve tomato quality to a certain extent.

Comprehensive analysis shows that aerated irrigation can promote tomato growth, improve tomato quality and increase the irrigation water utilization rate, and it can be applied to tomato processing by drip irrigation as a new technology. However, this experiment did not refine the aeration amount, did not observe the change in soil oxygen content and did not consider the aeration at different growth stages. Further research in this field will be carried out in the future.

**Author Contributions:** C.W.; methodology, software, investigation, data curation, formal analysis, data curation, conceptualization, writing—original draft preparation, visualization, Y.Z., J.Z., Z.W.; validation, resources, writing—review and editing, supervision, project administration, funding acquisition. All authors have read and agreed to the published version of the manuscript.

**Funding:** Shihezi University International Science and technology cooperation project "study on water demand characteristics and water and salt movement law of Processing Tomato under mulch drip irrigation in Northern Xinjiang" (GJHZ201803), 2019–2021.

**Institutional Review Board Statement:** Not applicable.

**Informed Consent Statement:** Not applicable.

**Data Availability Statement:** The data presented in this study are available on request from the corresponding author. The data are not publicly available due to privacy.

**Conflicts of Interest:** On behalf of all authors, the corresponding author states that there is no conflict of interest.

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
