# Peer review of "Evaluation of Suitable Mixture of Water and Air for Processing Tomato in Drip Irrigation in Xinjiang Oasis"

_sustainability, doi:10.3390/su13147845_

Round 1

Reviewer 1 Report

The development of new effective and water sawing irrigation technologies could contribute to achieving sustainable agriculture. The examined topic is actual and relevant but the manuscript has some major limitations that must be reconsidered.

  1. Indication of the year and the interval when the study was conducted would be required in line 87.
  2. Authors should add the flow rate and the amount of the applied irrigation water (m3/m2) to characterise the irrigation treatments.
  3. Authors mentioned that “In each plot, four seedling of processing tomato were sown” and even in Figure 3 four plants were presented. If this would be true, the number of plants would be too low to make scientifically appropriate observations. Please revise the number of plants planted to the experimental plots.
  4. Please indicate the main agronomic properties of the examined variety. Is it growth-determined or non-determined?
  5. Please take care of using the chemical symbols correctly: E.g. CO2.
  6. Authors should revise the sentence in line 208: “yield per plant, fruit 208 quality per plant and fruit quality.”
  7. Authors mentioned that “The quality of vitamin C, soluble solids, total sugars and total acids in processing tomatoes were determined at harvest time (August 22). Presumably, the quantity of these components was measured.
  8. The effect of two factors was tested in the manuscript, and the authors applied one-way ANOVA to determine the effects of the factors and “the interactions” on the measured variables. The one-way ANOVA is not suitable to describe interactions.
  9. The Results chapter starts with the interpretation of the soil moisture distribution but the authors did not indicate in the title that even the soil parameter was analysed in detail.
  10. Applying colour-coding in figures would be beneficial.
  11. Explanation the meanings of the indexes and error bars in the figure’s titles would be required.
  12. In Figure 6 (b) 7/5: A2W2 did not differ statistically from A2W1, but differed from A2W4?
  13. In Figure 8 (a) 7/23: Is the significant difference between A2W3 and A2W4 valid?
  14. Authors should take into consideration that if the interaction of the two factors is not significant the treatment combinations cannot be compared (A1W1 or A2W2). But if the effect of a single factor was significant the combination of those can be linked. E.g. Single fruit weight: Only the effect of the factor “A” was significant, therefore comparing A1W1and A2W1; A1W2 and A2W2 etc. would be relevant.
  15. Authors did not mention the correlation analysis and PCA in the M&M chapter.
  16. It must be taken into consideration that one of the basic criteria of the correlation analysis would be the independence of the correlated variables. E.g. the yield per plant and single fruit weight are not independent.

Reviewer 2 Report

The authors have evaluated how aerated irrigation influence the growth and production of tomatoes. The article is interesting, but I have found some flaws that require revisions.

The main concern is the insufficient analysis of the state of the art in the Introduction. The same problem for Discussions, where there are not cross-comparisons with other relevant studies.

In the abstract, the foreword and novelty of the study lack. Please report also shortly.

The quality of some figures is not acceptable, since reading them is hard.

English must be noticeably improved.

Some other concerns are reported in the commented MS in attachment.

Reviewer 3 Report

Dear Authors,

worldwide, the tomato is probably the most important horticultural species. Therefore, improving the yield and quality of this species is an important research issue. This manuscript presents interesting research on the effects of aerated irrigation on fruit processing tomato. I believe that the research is reliable, the results are interesting and the discussion is well structured. However, I have a few remarks to improve this manuscript:

1) In Abstract - there is no hypothesis.

2) Keywords - are a repetition of the words from the title of the manuscript, they should be different.

3) Figure 1. - graphics resolution is too low; geographic coordinates are unreadable.

4) Figure 2. - carelessly prepared.

5) Generally, Figures and Tables are sloppy; most graphs are illegible.

6) Both Tables and Figures should be understandable without additional reading of the text - all abbreviations should be explained in the header or footer.

7) In addition, the entire manuscript should be very carefully adapted to the Sustainability Template!

I believe that the manuscript represents remarkable research, but it still requires some technical and editorial refinement. After considering the above comments, I believe that the manuscript deserves to be published in the journal Sustainability. 

Round 2

Reviewer 1 Report

The manuscript is acceptable for publication in the present form.

Reviewer 2 Report

The authors have satisfactorily replied to all the reviewers' commenta and now the paper is significantly improved.